# The Function of BARD1 in Centrosome Regulation in Cooperation with BRCA1/OLA1/RACK1

**DOI:** 10.3390/genes11080842

**Published:** 2020-07-24

**Authors:** Kei Otsuka, Yuki Yoshino, Huicheng Qi, Natsuko Chiba

**Affiliations:** 1Department of Cancer Biology, Institute of Development, Aging and Cancer (IDAC), Tohoku University, 4-1 Seiryomachi Aoba-ku, Sendai 980-8575, Japan; kei.otsuka.e5@tohoku.ac.jp (K.O.); yuki.yoshino.c8@tohoku.ac.jp (Y.Y.); kaisei.sai.r2@dc.tohoku.ac.jp (H.Q.); 2Laboratory of Cancer Biology, Graduate School of Life Sciences, Tohoku University, 4-1 Seiryomachi Aoba-ku, Sendai 980-8575, Japan; 3Department of Cancer Biology, Tohoku University Graduate School of Medicine, 4-1 Seiryomachi Aoba-ku, Sendai 980-8575, Japan

**Keywords:** BARD1, BRCA1, tumor suppressor, centrosome, centriole duplication

## Abstract

Breast cancer gene 1 (BRCA1)-associated RING domain protein 1 (BARD1) forms a heterodimer with BRCA1, a tumor suppressor associated with hereditary breast and ovarian cancer. BRCA1/BARD1 functions in multiple cellular processes including DNA repair and centrosome regulation. Centrosomes are the major microtubule-organizing centers in animal cells and are critical for the formation of a bipolar mitotic spindle. BRCA1 and BARD1 localize to the centrosome during the cell cycle, and the BRCA1/BARD1 dimer ubiquitinates centrosomal proteins to regulate centrosome function. We identified Obg-like ATPase 1 (OLA1) and receptor for activated C kinase (RACK1) as BRCA1/BARD1-interating proteins that bind to BARD1 and BRCA1 and localize the centrosomes during the cell cycle. Cancer-derived variants of BRCA1, BARD1, OLA1, and RACK1 failed to interact, and aberrant expression of these proteins caused centrosome amplification due to centriole overduplication only in mammary tissue-derived cells. In S-G2 phase, the number of centrioles was higher in mammary tissue-derived cells than in cells from other tissues, suggesting their involvement in tissue-specific carcinogenesis by *BRCA1* and *BARD1* germline mutations. We described the function of BARD1 in centrosome regulation in cooperation with BRCA1/OLA1/RACK1, as well as the effect of their dysfunction on carcinogenesis.

## 1. Introduction

Germline mutations in *Breast Cancer gene 1* (*BRCA1*) are associated with familial breast and ovarian cancers [1]. In women with *BRCA1* mutations, the risk of developing breast cancer by the age of 70 years is 57% and that of ovarian cancer is 40% [2]. BRCA1 has a RING domain in the amino (N)-terminal region and two BRCT domains in the carboxy (C)-terminal region (Figure 1). BRCA1-associated RING domain 1 (BARD1) was identified as a binding protein of the N-terminal region (amino acid (aa) 1–304) of BRCA1 by yeast two-hybrid screening [3]. The *BARD1* gene, which maps to chromosome 2q35, is composed of 11 exons and encodes a protein of 777 aa [3]. BARD1 contains an N-terminal RING domain, three tandem ankyrin (ANK) repeats, and two BRCT domains (Figure 1). BARD1 forms a heterodimer with BRCA1 via their RING domain, and the C-terminal region of BRCA1 contributes markedly to the stability of the heterodimer [4,5]. The BRCA1/BARD1 dimer is involved in DNA repair, centrosome regulation, chromatin remodeling, and transcription [6].

Centrosomes are the major microtubule (MT)-organizing centers (MTOC) in animal cells; they control cell shape, polarity, and motility, and mediate the formation of a bipolar mitotic spindle [7,8]. Each centrosome consists of a pair of centrioles called the mother and daughter centrioles, surrounded by a protein matrix known as the pericentriolar matrix (PCM) (Figure 2A). The PCM contains γ-tubulin ring complexes (γ-TuRCs) that play important roles in nucleating, anchoring, and positioning MTs. The single centrosome in the G1 phase duplicates only once per cell cycle (in S phase), and one centrosome is inherited by each daughter cell [8]. Centrosome duplication is precisely controlled by centriole duplication during the cell cycle (Figure 2B). Centrosome duplication is initiated by the physical separation of a pair of centrioles (centriole disengagement) in late mitosis-early G1 phase. The new daughter centriole starts to form a procentriole perpendicular to each mother centriole in early S phase. Each daughter centriole gradually elongates during the S and G2 phases. In late G2 phase, the two centrosomes separate and migrate to form the two opposing poles of the mitotic spindle [9].

BARD1 localizes to the centrosome and functions in centrosome duplication and the regulation of MT organizing activity of centrosomes together with BRCA1 [10,11,12]. We recently identified the BRCA1/BARD1-interacting proteins Obg-like ATPase 1 (OLA1) and receptor for activated C kinase (RACK1) [13,14]. In this review, we focus on the function of BARD1 in centrosome regulation and its role as a tumor suppressor together with BRCA1/OLA1/RACK1. In addition, we describe the tissue-specific effects of dysregulation of these processes on carcinogenesis in breast cancer.

## 2. Centrosome Aberrations in Cancer

Alterations in centrosome number and structure often occur together and are associated with many cancers [15]. Centrosome aberration is detected in a broad range of solid and hematological malignancies, including breast cancer [16]. Several mechanisms can lead to numerical aberration and centrosome amplification (supernumerary centrosomes) (Figure 2C): (1) overduplication of centrioles, (2) cytokinesis failure followed by centrosome accumulation resulting from repeated centrosome duplication in successive cell cycles without cell division, (3) de novo synthesis of a centrosome, (4) mitotic slippage, and (5) cell fusion [16,17]. Centrosome amplification can result in chromosome segregation errors and abnormal cell division, leading to chromosomal instability (CIN) [18]. CIN is a major source of aneuploidy in cancer and associated with carcinogenesis and cancer progression.

Centrosome aberrations confer invasive properties that may lead to the formation of metastases through cell-autonomous and non-cell-autonomous mechanisms [19]. For example, centrosome amplification increases Rac1 activity, disrupting cell–cell adhesion and promoting cellular invasion [20]; it also promotes paracrine invasion by inducing the secretion of pro-invasive factors, including interleukin-8 [21]. Structural centrosome aberrations promote non-cell-autonomous dissemination of mitotic cells and extrusion of damaged cells from polarized epithelia, which are the initial steps of the metastatic process [22,23].

In breast cancer, centrosome aberrations are observed in the early stages of tumorigenesis [24,25,26] and are correlated with CIN [24,25,27]. Triple negative breast cancer (TNBC), a subtype characterized by negative expression of hormone receptors and lack of amplification/overexpression of human epidermal growth factor receptor type 2 (HER2), has high rates of recurrence, metastasis, and mortality [28]. Numerical and structural centrosome aberrations are more frequent in TNBC tissues than in non-TNBC tissues [27,29]. In breast cancer, centriole amplification is more frequent in hormone receptor-negative than in receptor-positive cell lines [30]. The majority of *BRCA1*-associated breast cancers belong to the TNBC subtype [31]. *BARD1* germline mutations are also observed in TNBC patients [32]. Furthermore, centrosome aberration is associated with adverse clinical factors and worse survival in patients with breast cancer [27].

*BRCA2* is another gene associated with hereditary breast cancer [33]. Centrosome aberration in breast cancer is associated with germline mutation of *BRCA1* or *BRCA2* and negative BRCA1 expression [34,35]. Centrosome amplification is observed even in the normal breast epithelium of *BRCA1* mutation carriers [36]. These findings suggest that centrosome aberrations occur in the early steps of tumorigenesis and are related to aggressive breast cancer features, and loss of BRCA1 function contributes to these processes.

## 3. The BRCA1/BARD1 Heterodimer Functions in Centrosome Regulation

BRCA1 and BARD1 localize to the centrosome throughout the cell cycle [11,12,37]. Two regions of BRCA1, aa 504–803 and aa 802–1002, mediate its binding to γ-tubulin [38,39]. Brodie et al. reported that both N and C-terminal regions of BRCA1, but not the RING domain, are required for its centrosomal localization, independently of BARD1 and γ-tubulin [40]. Tarapore et al. reported that only the middle portion of BRCA1, namely aa 802–1002, is responsible for its localization to centrosomes [39]. The N-terminal nuclear export sequence (NES) of BRCA1 is important for targeting, turnover, and function at the centrosome, suggesting regulation by chromosome region maintenance 1 (CRM1). In addition, the mitotic kinase Aurora A contributes to BRCA1 retention at the centrosome [40]. Inside the centrosome, BRCA1 localizes to mother centrioles, whereas daughter centrioles acquire BRCA1 prior to the initiation of procentriole formation in late G1 phase [39].

Similar to BRCA1, the N- and C-terminal regions of BARD1, but not the RING domain, are critical for its centrosomal localization independently of BRCA1 [41]. The N-terminal NES mediates the centrosomal localization of BARD1, suggesting that the CRM1 is also involved in this process. The RING domains are not necessary for the centrosomal localization of BRCA1 and BARD1. Fluorescence recovery after photobleaching assays indicate that the retained centrosomal pool of BARD1 is half the amount observed for BRCA1, and that BARD1 is one of the most highly mobile proteins in the centrosome [41].

The RING domains of BRCA1 and BARD1 have E3 ubiquitin ligase activity, which increases dramatically in the BRCA1/BARD1 RING domain heterodimer [42]. Several cancer-associated BRCA1 RING domain variants abolish binding to BARD1 and the E3 ubiquitin ligase activity [43,44]. The BRCA1/BARD1 dimer ubiquitinates centrosomal proteins, including γ-tubulin, nucleophosmin/B23 (NPM1), and receptor for hyaluronan (HA)-mediated motility (RHAMM)/hyaluronan-mediated motility receptor (HMMR) [10,45,46].

BRCA1/BARD1 monoubiquitinates γ-tubulin at lysines K48 and K344, and the C-terminal region of BRCA1 is required for this function [10]. Suppression or overexpression of BRCA1 or BARD1 results in centrosome amplification in mammary tissue-derived cells [10,47]. Centrosome amplification induced by BRCA1 suppression is caused by premature centriole disengagement and centriole reduplication [48]. These findings suggest that BRCA1 functions in the regulation of centrosome number by controlling centriole duplication in mammary cells. Furthermore, BRCA1/BARD1 inhibits centrosome-dependent MT organizing activity, and the C-terminal region of BARD1 is necessary for the inhibition [11,12]. MT organizing activity was analyzed by detecting aster formation by centrosomes the in vivo MT regrowth assay and in vitro assay. BRCA1/BARD1 E3 ubiquitin ligase activity and its inhibitory effect on MT aster formation are suppressed by Aurora A and promoted by protein phosphatase 1α [49]. Aurora A is a mitotic kinase, and its overexpression causes centrosome amplification in cells [50]. The E3 ubiquitin ligase activity of BRCA1/BARD1 is important for both functions, regulation of centriole duplication and the inhibitory effect on MT aster formation. Monoubiquitination of γ-tubulin at K344 is critical for both functions, whereas that at K48 functions only in centrosome duplication [11]. Partially consistent with these findings, embryonic fibroblasts, which are not mammary cells, from BRCA1-knockout mice, show centrosome amplification [51]. In MT organization by the centrosome, MT nucleation is initiated by the γTuRC, and then the MT anchoring complex at the sub-distal appendages in mother centrioles anchors the MT-nucleated γTuRC. The nucleated MTs then elongate to form MT asters. Terapore et al. analyzed MT nucleation and MT anchoring and/or elongation at the centrosome separately, and concluded that BRCA1 suppresses MT anchoring and/or elongation but not MT nucleation [39].

NPM1 interacts with the N-terminal region of BRCA1 and BARD1 in a manner dependent on BRCA1/BARD1 heterodimer formation and is polyubiquitinated by BRCA1/BARD1, resulting in its stabilization. In mitotic cells, NPM1 colocalizes with BARD1 at chromosomal surfaces and the perichromosomal cytoplasm, and with BRCA1 at the spindle poles [45].

RHAMM, a member of the transforming acidic coiled-coil (TACC) family, localizes to the centrosome and associates with MTs [52,53]. Similar to the effects of BRCA1 inhibition, depletion of RHAMM causes centrosome amplification in mammary tissue-derived cells. RHAMM is associated with BRCA1, BRCA2, and Aurora A, and is polyubiquitinated and stabilized by BRCA1/BARD1 [46]. The RHAMM ortholog in *Xenopus laevis*, XRHAMM, regulates spindle pole assembly mediated by the BRCA1/BARD1 heterodimer [54]. A yeast two-hybrid screening identified the association between the TAC-1 and BRD-1 *C. elegans* proteins, which are orthologs of TACCs and BARD1, respectively [55,56].

## 4. OLA1 and RACK1 Function in the Regulation of Centrosome Number Together with BRCA1/BARD1

Several cancer-derived mutations have been reported in the C-terminal region of BARD1 [57], and this region is required for BRCA1/BARD1-mediated inhibition of MT aster formation [12]. In a previous study investigating the function of BRCA1/BARD1, we identified OLA1 as an interacting protein with the C-terminal region of BARD1 (aa 546−777) using proteomics analysis [13]. OLA1 is a member of the Obg family and YchF subfamily of *P*-loop GTPases [58,59,60]. OLA1 is composed of a central guanine nucleotide-binding domain (G domain), flanked by a coiled-coil domain and a ThrRS-GTP-SpoT (TGS) domain [61] (Figure 3A), and has both ATPase and GTPase activity [61,62,63,64,65].

OLA1 directly binds to γ-tubulin, the N-terminal region of BRCA1 (aa 1−304), and the C-terminal region of BARD1 (Figure 3B). OLA1 interacts with the middle portion of BRCA1 via γ-tubulin and the C-terminal region of BRCA1. OLA1 localizes to the centrosome during the cell cycle (unpublished data). Similar to the effect of BRCA1 suppression, knockdown of OLA1 causes centrosome amplification via centriole overduplication, as well as the activation of aster formation by the centrosome. OLA1 with the E168Q variant, which is identified in breast cancer cell lines, fails to bind to the N-terminal region of BRCA1 and rescue OLA1 knockdown-induced centrosome amplification. The BRCA1 N-terminal I42V variant is proficient in a DNA double-strand break repair, homologous recombination [44], whereas it is deficient in the control of centrosome duplication [66]. The BRCA1 I42V variant abrogates BRCA1 binding to OLA1 [13] (Figure 3C), whereas the OLA1 E168Q variant inhibits centrosomal MT aster formation, similar to OLA1 wild-type. These findings suggest that binding of BRCA1 to OLA1 is critical for the regulation of centrosome duplication but not for centrosomal MT aster formation.

OLA is overexpressed in several malignancies [64]. Overexpression of OLA1 causes centrosome amplification via centriole overduplication. The OLA1 S36A, F127A, and T325A mutants fail to bind to the C-terminal region of BARD1 and rescue centrosome amplification induced by OLA1 knockdown. S36 and T325 are candidate phosphorylation sites, and the phosphomimetic S36D and T325E mutations are involved in the regulation of centrosome number and bind to the C-terminal region of BARD1 [47]. These findings suggest that OLA1 is first phosphorylated at S36 and T325 and then binds to the C-terminal region of BARD1. This is consistent with the BRCT domains in the C-terminal region of BARD1, which mediate binding to phosphorylated proteins [67]. The S36C variant, which is observed in cervical cancer, shows loss of the regulation of centrosome number and does not bind to the C-terminal region of BARD1. F127 is an ATP-binding residue [61] that is located near S36 in the tertiary structure of OLA1. T325 is located close to the binding surface of BARD1 when the structure of OLA1 is computationally docked to the C-terminal region of BARD1 [47].

Breast cancer-derived BARD1 variants C645R, V695L, and S761N [68,69] show decreased association with OLA1. The V695L variant shows loss of direct binding to OLA1 and decreased centrosomal localization and centrosome amplification by its overexpression; in addition, the variant fails to rescue the centrosome amplification induced by BARD1 knockdown in breast cancer cells. These results suggest that binding of BARD1 to OLA1 is also important for the regulation of centrosome number [47]. However, the V695L and S761N variants are proficient in homologous recombination [70,71]. Taken together, studies suggest that appropriate formation of the BRCA1/BARD1/OLA1 complex is necessary for the regulation of centrosome duplication (Figure 2C).

Work from our group identified RACK1 as an OLA1-interacting protein [14]. RACK1 is conserved from yeast to humans and is composed of seven Trp-Asp (WD) domains (Figure 2A). It functions as a scaffolding protein in multiple cellular processes [72]. RACK1 directly binds to BRCA1, OLA1, and γ-tubulin; it associates with BARD1 and localizes to centrosomes during the cell cycle. RACK1 is involved in the proper centrosomal localization of BRCA1. The cancer-associated BRCA1 variants R133H and E143K, and the RACK1 variant K280E, which decrease the binding of BRCA1 to RACK1, suppress the centrosomal localization of BRCA1. Furthermore, RACK1 is involved in centriole duplication, and its overexpression causes centrosome amplification via centriole overduplication in mammary tissue-derived cells [14].

## 5. Tissue-Specific Carcinogenesis Associated with Dysregulation of Centrosome Number Regulated by BRCA1/BARD1-Interacting Proteins

Aberrant expression of BRCA1, BARD1, OLA1, RACK1, or RHAMM causes centrosome amplification in mammary tissue-derived cells [10,13,14,46,47,48]. In cases of suppression of BRCA1 or OLA1 and overexpression of OLA1 or RACK1, centrosome amplification is due to centriole overduplication. Immunostaining the centrosome with anti-γ-tubulin antibody and the centriole with anti-centrin antibody showed that in mammary tissue-derived cells, most cells with two γ-tubulin spots contained more than two centrioles, whereas in other tissues, more than 50% of cells with two γ-tubulin spots contained only two centrioles [14]. This suggests that the number of centrioles is higher in mammary cells than in those derived from other tissues, which may be attributed to defects in the precise mechanisms controlling centriole duplications. Therefore, mammary cells may be sensitive to stimuli that cause centriole overduplication, such as alterations in the expression or mutations of BRCA1, BARD1, OLA1, RACK1, or RHAMM. This characteristic of mammary cells may play a role in tissue-specific carcinogenesis induced by *BRCA1* germline mutations, and could explain the high incidence of centrosomal amplification in aggressive breast cancers.

## 6. BARD1 Isoforms and Cancer

*BARD1* generates several transcripts by alternative splicing, which results in the deletion of the RING domain and/or ANK repeats (Figure 4). BARD1 was recently proposed to play a dual role in cancer [73,74]. Full-length BARD1 (FL-BARD1) functions as a tumor suppressor, whereas aberrant splice variants, BARD1 isoforms, play an oncogenic role. BARD1 isoforms are aberrantly expressed in various cancers, including breast cancer, and they are associated with poor prognosis [75,76,77,78,79,80,81,82,83]. Loss of FL-BARD1 is also correlated with poor prognosis [80].

The roles of BARD1 isoforms differ from those of FL-BARD1. BARD1β, which lacks the RING domain, localizes to the midbody during cytokinesis; it associates with and stabilizes Aurora B and BRCA2. By contrast, FL-BARD1 is involved in the ubiquitination and proteasomal degradation of Aurora B together with BRCA1 [84]. BARD1β suppresses homologous recombination repair in colon cancers [85].

The BARD1δ isoform, which lacks both the RING domain and ANK repeats, interacts with and stabilizes estrogen receptor (ER)α, antagonizing ERα ubiquitination by BRCA1/BARD1-FL [86]. Furthermore, BARD1δ suppresses the chromosome and telomere protective function of BRCA1/FL-BARD1 [87].

These BARD1 isoforms might affect the function of FL-BARD1 in the centrosome and cause centrosome amplification. BARD1β stabilize Aurora A in neuroblastoma [77], and neuroblastoma tissues show centrosome amplification [88]. The BARD1ω isoform, which encodes only ANK repeats and BRCT domains, is expressed at high levels and is associated with increased number of aberrant mitotic figures, such as aberrant chromosome alignment at metaphase and anaphase in leukemia cells [78]. Centrosome aberrations induced by overexpression of BARD1ω may be involved in these phenotypes.

## 7. Summary and Perspective

In this review, we described the function of BARD1 in centrosome regulation together with BRCA1. BRCA2, which interacts with BRCA1 and also localizes to the centrosome, plays a role in the regulation of centrosome duplication [89,90]. Germline mutations of *BARD1* are present in hereditary breast and ovarian cancers [69,91,92,93,94]. The role of BARD1, BRCA1, and BRCA2 in centrosome regulation may be important as tumor suppressors in hereditary breast cancer, as well as their function in DNA repair pathways. In addition, germline mutations of *BARD1* were recently reported in neuroblastoma [95,96], and centrosome aberrations may be involved in neuroblastoma development.

The functions of BARD1 in the centrosome are performed as E3 ubiquitin ligase forming a heterodimer with BRCA1. The BRCA1/BARD1-interacting proteins, OLA1 and RACK1 regulate centrosome duplication. RACK1 mediates the proper centrosomal localization of BRCA1 and regulates centriole duplication. However, the function of OLA1 in centrosome regulation remains unclear. OLA1 functions in centriole duplication and aster formation together with BRCA1/BARD1. OLA1 might also be involved in the ubiquitination of centrosomal proteins by BRCA1/BARD1.

In addition to BRCA1, BARD1, and BRCA2, DNA repair factors that localize to the centrosome and are involved in its regulation have been identified [97]. Deficiencies of centrosomal proteins affect the DNA damage response [98]. These findings indicate that centrosomal regulation is associated with the DNA damage response. OLA1 and RACK1 may also be involved in these processes, and their dysfunctions may play a role in carcinogenesis. Further investigation of the relationship and crosstalk between centrosomal regulation and the DNA damage response may provide insight into the mechanisms of carcinogenesis and tumor development.

## Figures and Tables

**Figure 1 genes-11-00842-f001:**
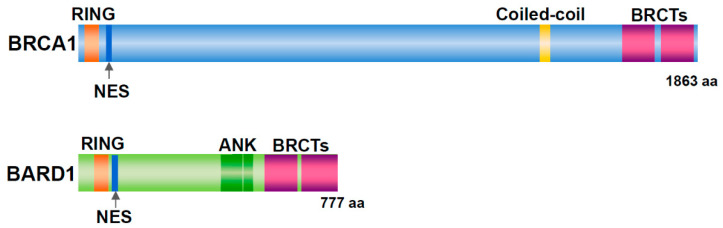
Structure of BRCA1 and BARD1. Both proteins have a RING domain and nuclear export signal (NES) in the N-terminal region and two BRCT domains in the C-terminal region. In addition, BRCA1 includes a coiled-coil domain. BARD1 contains three ankyrin (ANK) repeats.

**Figure 2 genes-11-00842-f002:**
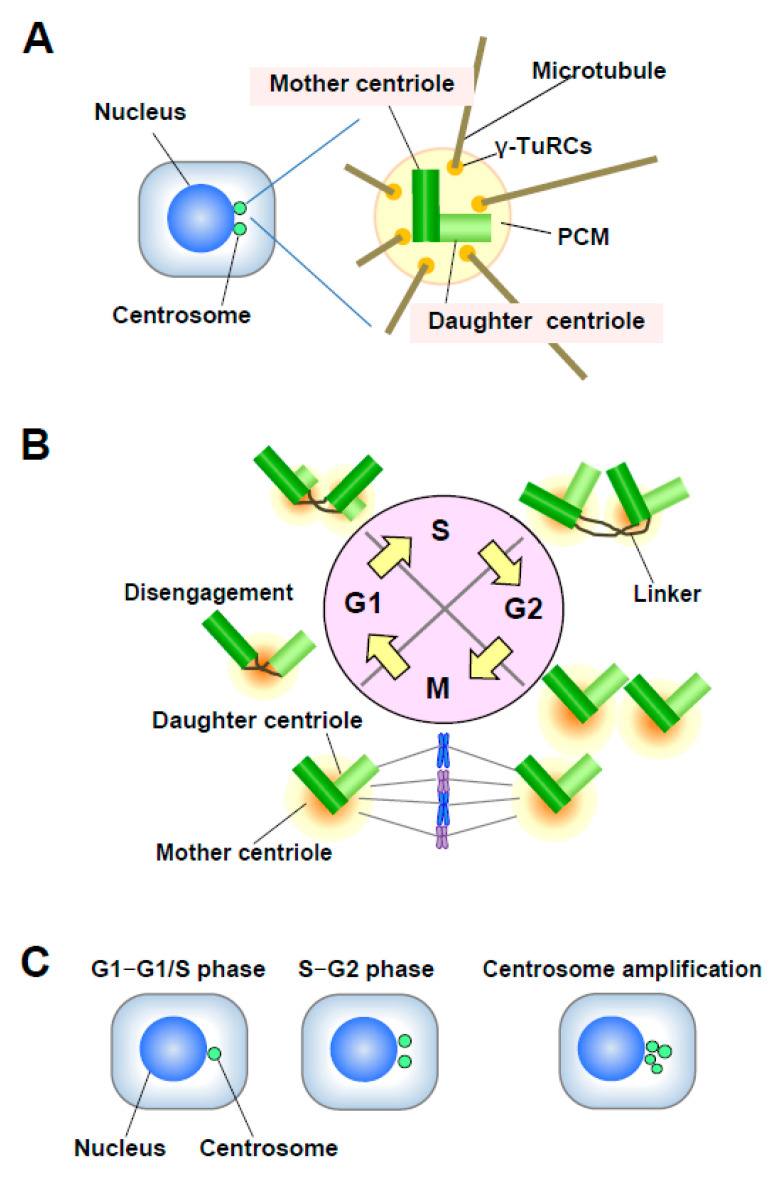
(**A**) Structure of the centrosome. The centrosome consists of a pair of centrioles, mother centriole and daughter centriole, embedded in the pericentriolar matrix (PCM). The PCM contains γ-TuRCs, which play roles in nucleating, anchoring, and positioning microtubules (MTs). (**B**) Centrosome duplication in the cell cycle. The mother and daughter centrioles are disengaged in late mitosis-early G1 phase. After centriole disengagement, a proteinaceous linker is established between the two centrioles and physically connects them. The building of the new centriole starts in the early S phase with the formation of a procentriole at each centriole. One new daughter centriole forms perpendicularly to each mother centriole during the S phase, and the new daughter centriole gradually elongates during the S and G2 phases. In late G2 phase, the two centrosomes separate through the dissolution of the linker and move to opposite sides of the cell to form the spindle poles. (**C**) Number of centrosomes. Normally, the centrosome number is one or two in interphase. Centrosome amplification is usually defined as more than two centrosomes per cell.

**Figure 3 genes-11-00842-f003:**
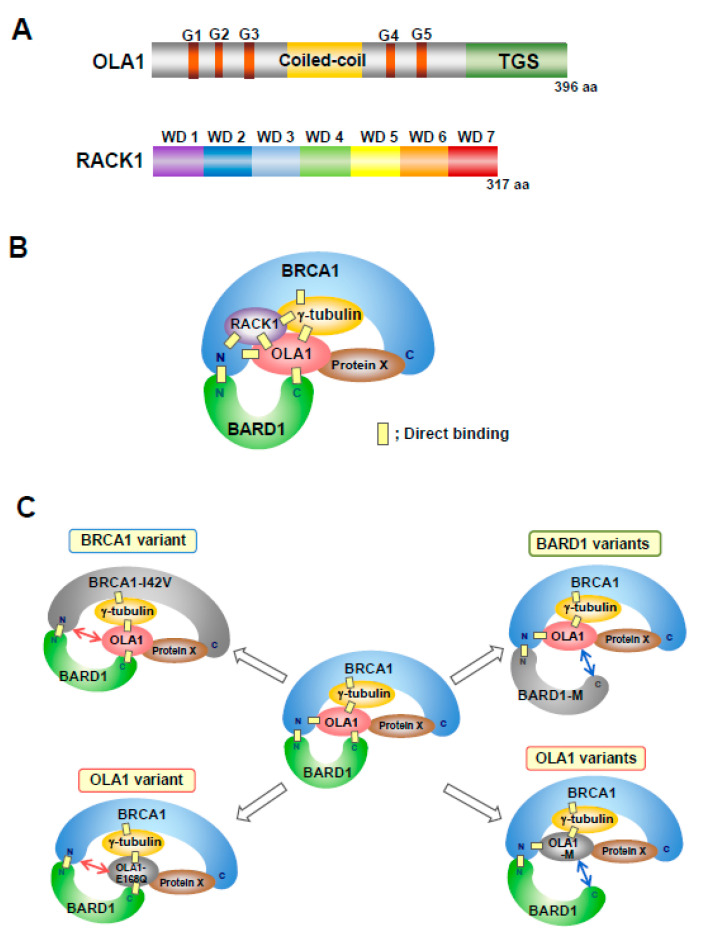
(**A**) Structure of Obg-like ATPase 1 (OLA1) and receptor for activated C kinase (RACK1). OLA1 is a member of the Obg family and YchF subfamily of *P*-loop GTPases, and is composed of a central guanine nucleotide-binding domain (G domain), flanked by a coiled-coil domain and a TGS domain. The G domain is the basic functional unit of GTP-binding proteins (G proteins) and contains five characteristic sequences (G1–G5) that are involved in nucleotide binding and hydrolysis. RACK1, a member of the Trp-Asp (WD) repeat protein family, is composed of seven WD domains that adopt *β*-propeller structures. (**B**) Model of the BRCA1/BARD1/OLA1/RACK1 complex. OLA1 binds to the N-terminal region of BRCA1, the C-terminal region of BARD1, and γ-tubulin. The N-terminal region of BRCA1 binds to the N-terminal region of BARD1. The middle portion of BRCA1 interacts with OLA1 via γ-tubulin. The C-terminal region of BRCA1 may be associated with OLA1 via an unknown protein, Protein X. RACK1 directly binds to OLA1, the N-terminal region of BRCA1, and γ-tubulin, and it is associated with BARD1. “N” indicates the N-terminal region. “C” indicates the C-terminal region. (**C**) Model of the conformational changes of the BRCA1/BARD1/OLA1/γ-tubulin complex induced by variants of BRCA1, BARD1, or OLA1. The N-terminal region of BRCA1 with the I42V variant shows markedly decreased binding to OLA1. OLA1 with the E168Q variant does not bind to the N-terminal region of BRCA1. These missense substitutions cause similar alterations in the conformation of the protein complex. The OLA1 mutations (M) S36A, S36C, F127A, and T325A abolish binding to the C-terminal region of BARD1. The BARD1 variants C645R, V695L, and S761N abolish binding to OLA1. These missense substitutions cause similar alterations in the conformation of the protein complex. “N” indicates the N-terminal region. “C” indicates the C-terminal region. These variants cause centrosome amplification. Abnormal formation of the complex is involved in carcinogenesis. BARD1-M; C645R, V695L, or S761N. OLA1-M; S36C, S36A, F127A, or T325A.

**Figure 4 genes-11-00842-f004:**
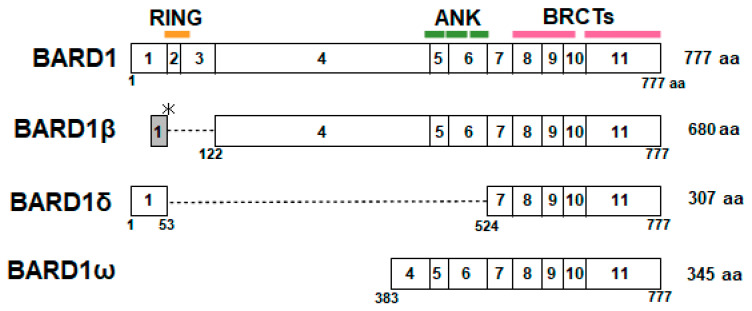
Structure of full-length (FL)-BARD1 and spliced isoforms of BARD1. BARD1β is translated from an alternative open reading frame (ORF) in exon 1, and exons 2 and 3 are deleted, resulting in the lack of the RING domain. In BARD1δ, exons 2–6 are deleted, and it lacks the RING domain and ANK repeats. BARD1ω consists of C-terminal region of exon 4 and exons 5–11, which encode the ANK repeats and BRCT domains.

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
