# Peer review of "The Function of BARD1 in Centrosome Regulation in Cooperation with BRCA1/OLA1/RACK1"

_genes, 2020, doi:10.3390/genes11080842_

Round 1

Reviewer 1 Report

The manuscript by Otsuka and colleagues provides a review of the role played by the complex BARD1/BRCA1/OLA1/RACK1 in centrosome regulation. The authors analyse the literature and report some published and unpublished data of their own work. This manuscript will make a timely contribution to the literature and provide a useful resource for researchers interested in the field.

However, some textual edits need to be made.

The authors should be careful in choosing specific terms such as “centrosome amplification, centrosome fragmentation and centrosome aberrations”.

Lines 81-82: The statement that “Centrosome amplification (>2 centrosomes), which is usually defined by immunostaining using antibodies against PCM proteins, such as γ-tubulin or pericentrin” is factually incorrect and conceptually flawed. Centrosome amplification could be assessed only by techniques that allow centrosome/centriole count. Staining with antibodies against PCM does not distinguish between centrosome fragmentation, with the generation of acentriolar MTOC and an increase in centrosome number. This is well defined in the cited review, page 1127: “The disintegration of centrosomes via fragmentation of PCM may lead to an appearance of CA”.

The authors make the same mistake in reporting literature. For example, line 109 ref [35]: Shimomura et al. reported data on centrosome aberration and not centrosome amplification found in breast cancer tissues.

I suggest that the authors edit the paragraph no. 2 paying attention to the appropriate use of the term “centrosome amplification”. In many cases it is better to use centrosome aberrations.

Line 142: “centriole disengagement (observed as fragmentation of the centrosome) and centriole overduplication (48)”. The reference is not properly cited: Ko et al. used GFP-centrin to distinguish between centriole disengagement and centriole overduplication. Furthermore, they did not describe any centrosome fragmentation in the cited paper.

Lane 171: “and this domain is required for MT nucleation induced by BRCA1/BARD1.” Wild type BARD1 domain inhibits microtubule nucleation from centrosome conversely the mutant domain promotes MT nucleation.

Therefore, the authors should clarify if they are talking of the role played by the wild type or mutant BARD1 domain.

Lane 249-250: “the number of centrioles is higher in mammary cells than in those derived from other tissues, which may be attributed to increased efficiency of centriole duplication or earlier timing of centriole duplication”. This sentence is not justified. The authors should elucidate why an increased efficiency of centriole duplication and earlier timing of centriole duplication can change the number of centrioles. As far as we know an increase in centrosome/centriole number is caused by defects in the precise mechanisms controlling centriole duplication and not by incorrect timing or efficiency of centriole duplication.

Finally, a suggestion: the title is not really appropriate because all the component of the complex BARD1/BRCA1/OLA1/RACK1 are equally discussed in this review, without any highlighting on BARD1.

Author Response

Reviewer #1 [reviewer comments in brackets]

[The manuscript by Otsuka and colleagues provides a review of the role played by the complex BARD1/BRCA1/OLA1/RACK1 in centrosome regulation. The authors analyse the literature and report some published and unpublished data of their own work. This manuscript will make a timely contribution to the literature and provide a useful resource for researchers interested in the field. However, some textual edits need to be made.]

     Thank you very much for the valuable comments. We have addressed your specific comments below.

[The authors should be careful in choosing specific terms such as “centrosome amplification, centrosome fragmentation and centrosome aberrations”.

Lines 81-82: The statement that “Centrosome amplification (>2 centrosomes), which is usually defined by immunostaining using antibodies against PCM proteins, such as γ-tubulin or pericentrin” is factually incorrect and conceptually flawed. Centrosome amplification could be assessed only by techniques that allow centrosome/centriole count. Staining with antibodies against PCM does not distinguish between centrosome fragmentation, with the generation of acentriolar MTOC and an increase in centrosome number. This is well defined in the cited review, page 1127: “The disintegration of centrosomes via fragmentation of PCM may lead to an appearance of CA”.]

    We agree with the reviewer. Our previous expressions were incorrect. In the revised manuscript, we removed the sentence, “which is usually defined by immunostaining using antibodies against PCM proteins, such as γ-tubulin or pericentrin” and changed the description of about the mechanism of centrosome amplification on page 4, lines 86-90 and the Figure legends of Figure 2C on page 3, lines 70-72 and reference 17 to Godinho SA et al. Cancer Metastasis Rev. 2009.

[The authors make the same mistake in reporting literature. For example, line 109 ref [35]: Shimomura et al. reported data on centrosome aberration and not centrosome amplification found in breast cancer tissues.]

    We agree with the reviewer. We changed “centrosome amplification” to “centrosome aberration” on page 4, line 112.

[I suggest that the authors edit the paragraph no. 2 paying attention to the appropriate use of the term “centrosome amplification”. In many cases it is better to use centrosome aberrations.]

     We agree with the reviewer. As the reviewer recommended, we changed “centrosome amplification” to “centrosome aberration” on page 4, line 83, 109, and 112.

[Line 142: “centriole disengagement (observed as fragmentation of the centrosome) and centriole overduplication (48)”. The reference is not properly cited: Ko et al. used GFP-centrin to distinguish between centriole disengagement and centriole overduplication. Furthermore, they did not describe any centrosome fragmentation in the cited paper.]

      We agree with the reviewer. In the revised manuscript, we changed “centriole disengagement (observed as fragmentation of the centrosome) and centriole overduplication” to “centriole disengagement and centriole reduplication” on page 5, lines 145-146.

[Lane 171: “and this domain is required for MT nucleation induced by BRCA1/BARD1.” Wild type BARD1 domain inhibits microtubule nucleation from centrosome conversely the mutant domain promotes MT nucleation.

Therefore, the authors should clarify if they are talking of the role played by the wild type or mutant BARD1 domain.]

     We agree with the reviewer. Previous description was incorrect. In the revised manuscript, we corrected the sentence to “this region is required for BRCA1/BARD1-mediated inhibition of MT aster formation.” on page 6, lines 179-180.

[Lane 249-250: “the number of centrioles is higher in mammary cells than in those derived from other tissues, which may be attributed to increased efficiency of centriole duplication or earlier timing of centriole duplication”. This sentence is not justified. The authors should elucidate why an increased efficiency of centriole duplication and earlier timing of centriole duplication can change the number of centrioles. As far as we know an increase in centrosome/centriole number is caused by defects in the precise mechanisms controlling centriole duplication and not by incorrect timing or efficiency of centriole duplication.]

    We agree with the reviewer. In the revised manuscript, we changed the sentence to “the number of centrioles is higher in mammary cells than in those derived from other tissues, which may be attributed to defects in the precise mechanisms controlling centriole duplication.” as the reviewer indicated on page 8, lines 260-261.

[Finally, a suggestion: the title is not really appropriate because all the component of the complex BARD1/BRCA1/OLA1/RACK1 are equally discussed in this review, without any highlighting on BARD1.]

     We agree with the reviewer. However, this review was prepared for special issue of “BARD1 in Cancer”. I would like to use this title, if possible.

Reviewer 2 Report

This review provides a summary of the conclusions from a large body of work surrounding the role of BARD1 and BRCA1 in centrosome regulation. Overall the manuscript is well written, provides a useful summary. However, the manuscript could be greatly improved if it provided additional value beyond a summary. For example it could add significant value to the field if it examined the limitations of the experiments that have been done, addressed inconsistencies and similarities among studies, or synthesized existing data into a unified model. In addition I have some minor concerns with the text.

  1. The authors should be careful to not state things seen in some circumstances (in one set of experiments, cell types, etc.) as general rules. For example, on lines 86 – 88, they state “Centrosome amplification results in chromosome segregation errors and abnormal cell division, leading to chromosomal instability (CIN) (18).” While this is something that can happen, centrosome amplification does not always lead to CIN. A more correct sentence would be “Centrosome amplification can result in chromosome segregation errors and abnormal cell division, leading to chromosomal instability (CIN) (18).”
  2. In summarizing the field, the conclusions of previous studies are taken at their word, even if the data from the studies might not support those conclusions. Addressing these kind of issues would be an incredible value to the community and help prevent the propagation of poor conclusions. I cite two examples.
    1. There is a lot of discussion of “microtubule (MT) nucleation” throughout this manuscript. While many of the previous studies conclude that the effects they are seeing on centrosomes are due to effects on MT nucleation, the assay (looking for asters in MT regrowth assay) used in many doesn’t only assay nucleation. In fact in this manuscript, they highlight a study by Tarapore that “analyzed MT nucleation and MT anchoring and/or elongation at the centrosome separately, and concluded that BRCA1 suppresses MT anchoring and/or elongation but not MT nucleation” using assays developed by Bornes. It is very possible that some of the other studies that concluded an effect on MT nucleation, were actually doing an assay that couldn’t discriminate between nucleation and anchoring. It is also possible that I am not aware of studies that use other techniques to support a conclusion of an effect on nucleation (rate of EB1 emission from the centrosome for example). It would be of great service to the field if the potential for some previous results being an effect on something other than nucleation was discussed, rather than relying on the conclusions of the original papers.
    2. A second example of where the previous authors are taken at their word is on lines 127 – 128. “The RING domains are not necessary for the centrosomal localization of BRCA1 and BARD1, indicating that BRCA1 and BARD1 localize to the centrosome prior to heterodimerization.” But when I look at the experiments in this paper, they show that BRCA1 and BARD do not depend on each other to localize. These experiments do not provide any evidence as to when they heterodimerize.

Author Response

Reviewer 2 [reviewer comments in brackets]

[This review provides a summary of the conclusions from a large body of work surrounding the role of BARD1 and BRCA1 in centrosome regulation. Overall the manuscript is well written, provides a useful summary. However, the manuscript could be greatly improved if it provided additional value beyond a summary. For example it could add significant value to the field if it examined the limitations of the experiments that have been done, addressed inconsistencies and similarities among studies, or synthesized existing data into a unified model. In addition I have some minor concerns with the text.]

    Thank you very much for your really valuable comments. We agree with the reviewer. However, in this manuscript, we changed only parts that you pointed out as below.

[1.The authors should be careful to not state things seen in some circumstances (in one set of experiments, cell types, etc.) as general rules. For example, on lines 86 – 88, they state “Centrosome amplification results in chromosome segregation errors and abnormal cell division, leading to chromosomal instability (CIN) (18).” While this is something that can happen, centrosome amplification does not always lead to CIN. A more correct sentence would be “Centrosome amplification can result in chromosome segregation errors and abnormal cell division, leading to chromosomal instability (CIN) (18).”]

     We agree with the reviewer. We have changed ‘‘Centrosome amplification results in chromosome segregation errors and abnormal cell division, leading to chromosomal instability (CIN)’’ to ‘‘Centrosome amplification can result in chromosome segregation errors and abnormal cell division, leading to chromosomal instability (CIN) (18)", as the reviewer recommended on page 4, line 91.

[2.In summarizing the field, the conclusions of previous studies are taken at their word, even if the data from the studies might not support those conclusions. Addressing these kind of issues would be an incredible value to the community and help prevent the propagation of poor conclusions. I cite two examples.

1.There is a lot of discussion of “microtubule (MT) nucleation” throughout this manuscript. While many of the previous studies conclude that the effects they are seeing on centrosomes are due to effects on MT nucleation, the assay (looking for asters in MT regrowth assay) used in many doesn’t only assay nucleation. In fact in this manuscript, they highlight a study by Tarapore that “analyzed MT nucleation and MT anchoring and/or elongation at the centrosome separately, and concluded that BRCA1 suppresses MT anchoring and/or elongation but not MT nucleation” using assays developed by Bornes. It is very possible that some of the other studies that concluded an effect on MT nucleation, were actually doing an assay that couldn’t discriminate between nucleation and anchoring. It is also possible that I am not aware of studies that use other techniques to support a conclusion of an effect on nucleation (rate of EB1 emission from the centrosome for example). It would be of great service to the field if the potential for some previous results being an effect on something other than nucleation was discussed, rather than relying on the conclusions of the original papers.]

    We agree with the reviewer. In the revised manuscript, we changed “MT nucleation” to “MT organizing activity of centrosomes” on page 3, line 75 and page 5, lines 148-150 and to “MT aster formation” on page 5, lines 152 and 156. And we corrected the description in “aster formation by centrosomes in vitro and in vivo” to “aster formation by centrosomes in vivo MT regrowth assay and in vitro assay” on page 5, lines 150 and 151. Furthermore, we added the description about MT organization by centrosome, “In MT organization by centrosome, MT nucleation is initiated by the γTuRC, and then the MT anchoring complex at the sub-distal appendages in mother centrioles anchors the MT-nucleated γTuRC. The nucleated MTs then elongate to form MT asters.” on page 5, lines 159-161.

[2.A second example of where the previous authors are taken at their word is on lines 127 – 128. “The RING domains are not necessary for the centrosomal localization of BRCA1 and BARD1, indicating that BRCA1 and BARD1 localize to the centrosome prior to heterodimerization.” But when I look at the experiments in this paper, they show that BRCA1 and BARD do not depend on each other to localize. These experiments do not provide any evidence as to when they heterodimerize.]

      We agree with the reviewer. We removed the sentence of ‘‘indicating that BRCA1 and BARD1 localize to the centrosome prior to heterodimerization’’ on page 5, line 132.